

# Design of performance evaluation method for higher education reform based on adaptive fuzzy algorithm

Dakun Yang[1] and Muhammad Sheraz Arshad Malik[2]

[1] School of Marxism, Anyang Institute of Technology, Anyang, China
[2] Department of Software Engineering, Government College University, Faisalabad, Faisalabad, Pakistan

## ABSTRACT

This study presents a performance evaluation framework for university teachers based on the adaptive neural fuzzy inference system (ANFIS), aiming to enhance teaching quality and institutional management through a scientific, objective, and comprehensive assessment mechanism. The proposed method begins by developing a robust evaluation index system that integrates key dimensions of academic activity, including teaching performance, research contributions, and fundamental faculty information. A total of 16 sub-indicators are incorporated into the evaluation framework. To optimize data processing and reduce redundancy, factor analysis is applied, simplifying the indicator set while maintaining the integrity and effectiveness of the evaluation process. The core of the system leverages the strengths of both fuzzy logic and neural networks, combining the capacity of fuzzy systems to handle imprecise and uncertain information with the adaptive learning capabilities of neural networks. This hybrid approach improves the accuracy, interpretability, and adaptability of the evaluation results. By continuously optimizing the model using training data, the system dynamically refines its rule base and parameters, eliminating the reliance on manually defined parameters common in traditional fuzzy systems. The effectiveness of the ANFIS-based evaluation model is validated through empirical experiments. The results demonstrate that the proposed model outperforms conventional methods, such as backpropagation (BP) neural networks and support vector machines (SVMs), in terms of accuracy, precision, and overall performance. This research offers a novel and practical approach for evaluating university teacher performance, enabling more accurate reflection of teaching and research outcomes, and providing valuable decision-making support for academic management.

# INTRODUCTION

In the construction of universities, the construction of the teaching staff plays a decisive role in the quality of teaching, which directly affects the level of education and the quality of student training (*Hussain et al., 2023*; *Abdallah & Abdallah, 2023*). To improve the quality and level of teaching, universities conduct comprehensive annual assessments and evaluations of teachers, aiming to assess their pedagogical skills, theoretical teaching

Corresponding author
Dakun Yang, 20160842@ayit.edu.cn

abilities, practical teaching skills, and overall performance. However, evaluating the teaching quality of teachers scientifically and reasonably, promoting teaching reform, and stimulating teachers' teaching enthusiasm to improve the overall teaching level and student training quality of schools, has always been an essential issue in university teaching management.

Teaching quality evaluation is a broad and complex task that is influenced and constrained by various factors and conditions (*Chia et al., 2023*). There are problems, such as insufficient consensus and poor comparability, in the existing evaluation system, which necessitate improvements in the scientificity and rationality of the evaluation work. Therefore, how to scientifically and quantitatively evaluate the quality of teaching so that the evaluation results can be truly fair, reasonable, and objective, and to accurately assess each teacher, has become an important topic that urgently needs to be studied. With the widespread application of computer and information network technology, these technologies have penetrated various aspects of social life. The traditional evaluation and assessment system, based on subjective feelings, has gradually exposed its limitations due to human interference and therefore urgently needs reform. Using computer software systems for evaluation is an essential objective evaluation method that can significantly reduce the influence of human factors and improve the objectivity and reliability of evaluation results (*Zhang, 2023*).

However, in the actual evaluation process, there is no absolute precision standard, which brings new challenges to scientific evaluation. To solve this problem, we combined fuzzy mathematics theory to design and implement a university teacher evaluation system based on a fuzzy evaluation algorithm. This system can comprehensively handle the fuzziness and uncertainty in evaluation, providing more reasonable and scientific evaluation results, and offering strong support for assessing university teachers. Artificial neural networks and fuzzy logic have been widely applied in the research of various pattern recognition problems. Artificial neural network systems are essentially models that simulate the processing of information by the human brain, but they cannot handle fuzzy details (*Luo et al., 2023*; *Croix, Jean & Ahmad, 2023*). Fuzzy logic systems can express human experiential knowledge and are adept at handling fuzzy information. Still, their rule sets and membership functions often rely on empirical selection and are difficult to adjust automatically. This is precisely the advantage of neural networks. By combining neural networks with fuzzy logic, a fuzzy neural system can compensate for its respective shortcomings and provide an efficient solution for network learning, evaluation, and modeling.

This article proposes an adaptive fuzzy neural network system aimed at improving the accuracy of performance evaluation for university teachers. The specific contributions are as follows: This article combines the advantages of fuzzy logic and neural networks to design and implement an adaptive fuzzy neural network system that can effectively handle the fuzziness and uncertainty in teacher performance evaluation. At the same time, the system parameters are optimized through network learning to improve the accuracy and fairness of the review.

Compared with other methods, beyond accuracy, adaptive neural fuzzy system (ANFIS) offers several advantages for educational performance evaluation. It combines the learning ability of neural networks with the interpretability of fuzzy logic, enabling the integration of expert knowledge and the handling of uncertainty. This makes ANFIS especially suitable for decision-making contexts, such as faculty assessment, where transparency and stakeholder trust are crucial. Additionally, ANFIS performs well on small to medium-sized datasets and provides human-readable fuzzy rules, making it more practical than black-box models such as SVMs or deep learning.

## RELATED WORKS

With the deepening of educational informatization and the continuous enrichment of teaching resources, the evaluation of teaching quality and performance in universities has gradually become an essential part of academic management. In recent years, universities have achieved significant progress in developing teaching resources, including the optimization of curriculum systems and the upgrading of teaching platforms. However, there is still a phenomenon of "emphasizing construction over evaluation", which affects the deep promotion of teaching reform and the comprehensive improvement of resource efficiency. Constructing a performance evaluation system for universities is relatively straightforward. Still, it is challenging to implement in practice, mainly due to the lack of a scientific and comprehensive performance evaluation mechanism. Therefore, establishing a scientific and efficient performance evaluation system can not only accurately reflect teaching achievements but also promote the effective utilization of teaching resources, optimize teaching management, and improve the overall level of higher education. At present, commonly used learning evaluation methods include the expert evaluation method (*Yakhyaeva & Skokova, 2021*), the fuzzy comprehensive evaluation method (*Ting, 2021*), and the neural network evaluation method (*Wang et al., 2023*).

### Expert evaluation method

The expert evaluation method is a method based on the theoretical knowledge and practical experience of experienced experts, widely used in the fields of university teacher performance evaluation, education quality evaluation, and scientific research achievement evaluation (*Tao et al., 2023*). The main feature of this method is that it relies on the subjective judgment of experts and conducts a comprehensive analysis of the evaluation object through professional knowledge and industry standards. This method possesses a certain degree of authority and reliability, making it especially suitable for complex evaluation tasks that combine qualitative and quantitative approaches (*Mondal, Roy & Zhan, 2023*).

*Anderson & Taner (2023)* analyzed 106 empirical studies from 16 countries and systematically summarized the professional knowledge of K12 education expert teachers for the first time. He found that expert teachers exhibit characteristics such as critical reflection, continuous learning, and building strong relationships with students in teaching practice, and possess rich teaching content and in-depth knowledge of learners. *Ruiz, Segura & Sirvent (2015)* extended the data envelopment analysis (DEA) model to

incorporate expert preferences into the analysis, providing the closest goals and applying them to the educational performance evaluation of public universities in Spain to optimize goal setting and improve actual performance. However, the expert evaluation method also has certain limitations. On the one hand, due to the high dependence of the evaluation process on the subjective judgment of individual experts, differences in cognition and experience among different experts may lead to inconsistencies in the evaluation results. On the other hand, when there are a large number of evaluation objects or a complex indicator system, expert evaluation is prone to low efficiency due to the large workload. Additionally, the authority of expert opinions may raise concerns about individual bias or group bias, thereby reducing the objectivity of evaluations (*Alrakhawi et al., 2024*).

## Fuzzy comprehensive evaluation method

Fuzzy comprehensive evaluation (FCE) is a comprehensive evaluation technique that combines fuzzy mathematical theory with multiple indicator evaluation methods. *Xu et al. (2023)* This method can handle complex problems that are difficult to handle with traditional quantitative analysis methods due to incomplete information and uncertain or ambiguous evaluation objects. It has therefore been widely applied in various fields, including the environment, economy, and management.

*Wang & Yang (2023)* proposed a university teaching quality evaluation model based on a fuzzy comprehensive evaluation method. By constructing a multidimensional indicator system that encompasses teacher resources, teaching environment, and teaching effectiveness, researchers utilize fuzzy logic to assign weights to each indicator and comprehensively analyze the evaluation results based on expert opinions. *Chen, Hsieh & Do (2015)* proposed a performance evaluation model for university scientific research based on a fuzzy comprehensive evaluation method. By analyzing multiple dimensions of scientific research activities in universities, such as research projects, research funding, article publication, and research achievement transformation, researchers employ fuzzy mathematical methods to comprehensively evaluate these indicators. The results indicate that the fuzzy comprehensive evaluation method can better handle the combination of qualitative and quantitative data in scientific research performance evaluation, thereby improving the accuracy and effectiveness of the evaluation. Apply the fuzzy comprehensive evaluation method to the assessment of financial performance in universities. *Ying & Zhi (2024)* applied the fuzzy comprehensive evaluation method to the financial performance evaluation of universities.

## Neural network evaluation method

Artificial neural networks and fuzzy logic have been widely applied in the research of various pattern recognition problems. Artificial neural network systems are essentially models that simulate the processing of information by the human brain, but they cannot handle fuzzy details. Fuzzy logic systems can express human experiential knowledge and are adept at handling fuzzy information. Still, their rule sets and membership functions often rely on empirical selection and are difficult to adjust automatically. This is precisely the advantage of neural networks. By combining neural networks with fuzzy logic, a fuzzy

neural system can compensate for its respective shortcomings and provide an efficient solution for network learning, evaluation, and modeling (*Vargas et al., 2023*; *Gupta, Sethi & Goswami, 2024*).

*Bishop & Nasrabadi (2006)* systematically introduced neural network methods in pattern recognition and machine learning, including multi-layer perceptrons and deep learning algorithms. Although this book does not explicitly explore performance evaluation in universities, its content provides a theoretical basis for understanding the application of neural networks in university performance evaluation. *Zhang et al. (2021)* proposed a neural network-based performance evaluation model for higher education. The researchers employed a backpropagation neural network (BPNN) to comprehensively evaluate multiple indicators, including educational resources, teaching quality, and research output, in universities. By learning and training on a large amount of historical data, the model can accurately reflect the educational performance of universities and update its evaluation results in real-time. *Yu (2024)* used deep learning models to evaluate the quality of teaching in universities. Researchers utilize deep neural networks (DNNs) to train and evaluate data across multiple dimensions, including teacher evaluation, student satisfaction, course content, and other relevant factors. The results show that deep learning models can extract useful features from a large amount of complex teaching data, providing a more accurate evaluation of teaching quality.

### International rating systems for teacher effectiveness

Leading universities worldwide employ diverse evaluation frameworks. For instance, the UK's Teaching Excellence Framework (TEF) combines student satisfaction, learning outcomes, and employment data (*Gunn, 2018*), whereas the U.S. Faculty Course Evaluations (FCEs) emphasize teaching quality and curriculum design (*Constantinou & Wijnen-Meijer, 2022*). Compared to these traditional methods, ANFIS offers advantages in handling multidimensional and fuzzy indicators (*e.g.*, research impact, student feedback) through data-driven rule generation.

## METHODOLOGY

### Establishment of performance evaluation system for higher education reform based on adaptive neural fuzzy system

*Overall design framework of the evaluation system*

The overall design framework of the evaluation system is shown in Fig. 1. Figure 1 illustrates the overall design framework of the evaluation system, which includes factor analysis and the design process of the adaptive neural fuzzy system. Based on existing research results and combined with the requirements of scientificity, accuracy, and the convenience of practical application in enterprises, the evaluation system design framework of this article is divided into two parts:

Factor analysis and simplified indicator system: Firstly, factor analysis is used to extract factors from the current complex and diverse indicator system, constructing a more concise and clear evaluation framework. This approach not only refines the meaning of

**Peer**J Computer Science

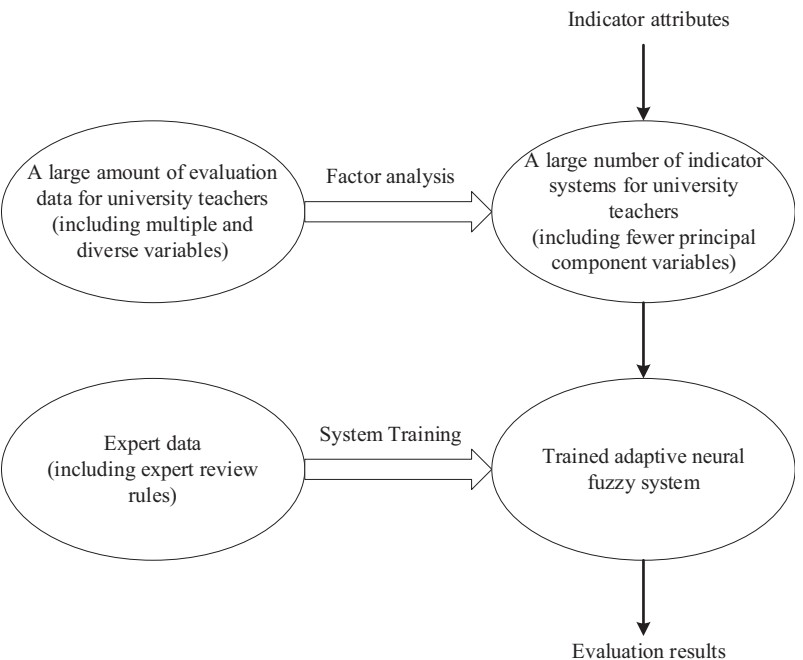

**Figure 1 Overall design framework of evaluation system.**

indicators but also effectively reduces data redundancy, laying an efficient foundation for the subsequent computation of adaptive neural fuzzy systems.

Design of an adaptive neural fuzzy system: Secondly, by combining fuzzy theory and neural network theory, design an adaptive neural fuzzy system. By introducing expert data for system training, the system can automatically generate scientifically accurate rules, achieving intelligence in supplier evaluation and selection.

### Performance evaluation system for higher education reform based on factor analysis

There have been numerous achievements in research on the performance evaluation of education reform, both domestically and internationally, typically involving key indicators such as teaching management levels, teacher team construction, student development effectiveness, and resource allocation efficiency. Based on the characteristics of the current new stage of higher education reform, this article has added two indicators: student innovation ability cultivation and educational informatization level. By drawing on existing research, a performance evaluation system consisting of 16 sub-indicators has been constructed.

To refine the meaning of indicators and avoid computational redundancy caused by data processing complexity, this article employs the factor analysis method to optimize and reconstruct the performance evaluation indicator system, extract the main factors, and establish a more concise and clear evaluation framework. This design will provide a solid foundation for future performance evaluation based on adaptive fuzzy algorithms.

$$X_i = a_{i1}F_1 + a_{i2}F_2 + \cdots + a_{im}F_m + \varepsilon_i, i = 1, 2, \cdots, 16 \tag{1}$$

where $X_i$ refers to the i-th sub indicator, and $F_m$ refers to the m extracted principal components. Equation (1) represents the relationship between the principal components extracted through factor analysis and the original indicators.

Through factor analysis and variance contribution rate, j principal components $(1 \leq j \leq m)$ can be selected, and the score calculation formula for each factor can be obtained:

$$F_j = b_{j1}X_1 + b_{j2}X_2 + \cdots + b_{j16}X_{16}, 1 \leq j \leq m. \tag{2}$$

This achieves the simplification of the indicator system $(m - j)$ and provides a few variable dimensions for subsequent model calculations; In addition, by explaining the meaning of $F_j$, the meanings of each indicator can be more refined, thereby providing a more intuitive understanding of educational evaluation based on this foundation.

## Adaptive network-based fuzzy inference system model

The adaptive neuro-fuzzy inference system (ANFIS) is a system that combines fuzzy theory with neural networks to achieve inference and learning through specific algorithms. In recent years, both fuzzy logic and neural networks have made significant progress in theory and application; however, they still have certain limitations. First, although fuzzy reasoning systems can effectively handle fuzzy language and uncertainty, they cannot self-learn. They are difficult to adapt to dynamic and complex application scenarios, limiting their popularity and use. Secondly, although artificial neural networks possess strong learning and nonlinear mapping capabilities, they struggle to handle fuzzy language in the design process, and their internal computational processes lack transparency, making it challenging to express logical mechanisms similar to human reasoning accurately.

ANFIS combines the advantages of fuzzy logic and neural networks organically, utilizing the learning mechanism of neural networks to compensate for the insufficient learning ability of traditional fuzzy systems, while retaining the interpretability of fuzzy language. This combination enables ANFIS to perform well in handling complex nonlinear problems, fuzzy language modeling, and data-driven learning scenarios, with broad application potential.

Figure 2 is a schematic diagram of the five-layer structure of the adaptive neural fuzzy system, corresponding to the fuzzification, fuzzy operation, normalization, rule output calculation, and total output of the input data:

First layer: Implement fuzzification of input data.

When data x containing j variables is input into the system, the data is first fuzzified using membership function processing.

The membership function is a mathematical tool for representing fuzzy sets, with values in the interval [0, 1] indicating the degree to which each element belongs to the fuzzy set. Standard membership functions include triangular, trapezoidal, bell-shaped, and Gaussian membership functions. The Gaussian membership function expression used in this article is:

$$u(x) = e^{-\frac{(x-c)^2}{2\sigma^2}}. \tag{3}$$

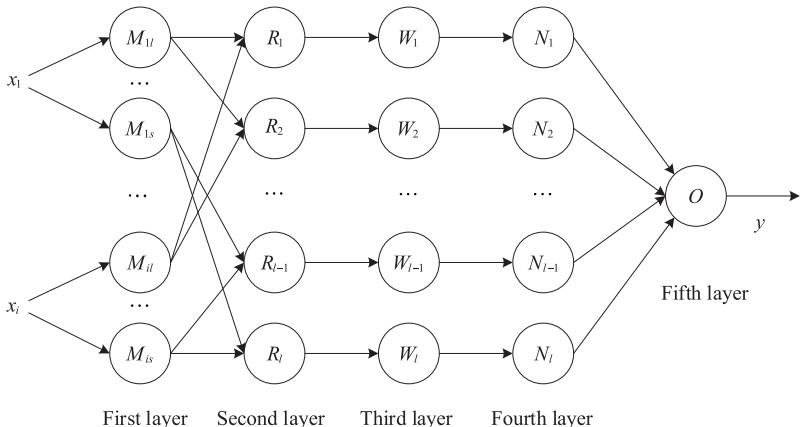

**Figure 2 Structure diagram of adaptive neural fuzzy system.**

The parameters c and $\sigma$ in the Gaussian membership function represent the center and width of the membership function, respectively, and their values are adaptively adjusted through training data.

Second layer: Implement fuzzy data operations.

The fuzzy operation of this layer can be expressed in logical language as follows:

$R^{(1)}$ : if $x_1$ is $M_{1s}$, $x_2$ is $M_{2t}$, $\cdots$, $x_j$ is $M_{jn}$; then $y$ is $R^{(1)}$, where $M_{ij}$ refers to the j-th membership function corresponding to the i-th variable, and the total number of rules d in the system can be obtained from the above rules:

$$d = \prod_{i=1}^{j} m_i \tag{4}$$

where j refers to the number of variables in the data, and $m_i$ is the number of membership functions corresponding to the i-th variable. Specifically, when the number of membership functions corresponding to each variable is the same and all are m, $d = m_j$.

There can be various algorithms, such as algebraic product, taking the small, bounded product, *etc.* This article adopts the algebraic product form:

$$R^{(1)} = w_j = u_{1s}(x_1)u_{2t}(x_2) \cdots u_{jn}(x_j) \tag{5}$$

where $u_{jn}(x_j)$ refers to the function value of the j-th variable corresponding to the n-th membership function. $w_l$ can also be referred to as the incentive strength of the rule.

Third layer: Normalization of incentive intensity:

$$W_l = \bar{w}_l = \frac{w_l}{\sum_{i=1}^{d} w_i}. \tag{6}$$

Fourth layer: Implement the rule output result calculation

$$N_l = y_l = \bar{w}_l f_1 = \bar{w}_l (p_{1s}x_1 + p_{2t}x_2 + \cdots + p_{jn}x_j + r_l) \tag{7}$$

where $p_{1s}, \cdots, p_{jn}$ and $r_l$ are the consequent parameters.

Fifth layer: Implement total output

$$O = \sum_{l=1}^{d} y_l = \sum_{l=1}^{d} \bar{w}_l \left( p_{1s} x_1 + p_{2t} x_2 + \cdots + p_{jn} x_j + r_l \right). \tag{8}$$

In the above system, some factors need to be determined, including the types and quantities of membership functions, antecedent parameters, and consequent parameters. The types and quantities of membership functions are inherent properties of the system and can be determined through experience and subsequent trial-and-error methods. The antecedent and consequent parameters need to be trained on the system using training data that contains expert information.

In the training of antecedent and consequent parameters, hybrid learning algorithms are usually used. For the antecedent parameters, the backpropagation algorithm is used to learn and adjust, while for the consequent parameters, the least squares algorithm is used for adjustment. In each iteration, the input signal first propagates forward along the system network until the fourth layer, where the antecedent parameters are fixed and the consequent parameters are adjusted using the least squares estimation algorithm; Afterwards, the signal continues to propagate forward along the network until it reaches the input layer. Afterwards, the error signal of the output result is used to propagate back along the system optical network, thereby adjusting the antecedent parameters. Repeatedly iterating in this way, all parameters are finally trained.

To extract the most representative sub-indicators for the evaluation model, we applied exploratory factor analysis (EFA) using principal component extraction and varimax rotation. Before conducting the EFA, we assessed sampling adequacy using the Kaiser-Meyer-Olkin (KMO) test, which yielded a value of 0.813, indicating adequate sampling. Bartlett's test of sphericity was also significant ($p < 0.001$), confirming the suitability of the data for factor analysis.

Sub-indicators with factor loadings above 0.6 and communalities greater than 0.5 were retained for further analysis. The extracted factors accounted for 76.4% of the total variance. After statistical filtering, the remaining sub-indicators were reviewed by three senior experts in education management to ensure domain relevance and consistency with the theoretical model. This combined statistical and expert-driven approach ensured the scientific rigor and practical applicability of the selected indicators.

The reason for choosing EFA is that it can effectively handle redundancy issues in high-dimensional data and simplify the indicator system through principal component extraction. The results of EFA provide ANFIS with more concise input variables, thereby reducing model complexity and improving computational efficiency. In addition, the variance contribution analysis of EFA ensures that the extracted principal components can fully represent the variability of the original data.

## Parameter adjustment

Numerical adjustment includes initialization of network parameters (here, the error accuracy is set to 0.01, the maximum training times are 500, and other parameters are initialized with random values) and parameter training. The process of parameter

**Table 1 Determination method of corresponding variables.**

| Variable types | Variable description variable description | Determination method |
|---|---|---|
| Types of membership functions. | The system itself constitutes attributes. | Determine through experience and trial and error methods. |
| Front piece parameters. | Mainly membership function parameters, such as Gaussian functions $c$ and $\sigma$. | Training determination, mainly the backpropagation algorithm. |
| Rear part parameters. | Mainly the parameters in the output result calculation formula. | Training determination mainly involves the least squares algorithm. |

adjustment is a continuous training and testing process. When the detection accuracy reaches a predetermined value (allowable error value) or the training time reaches its maximum value, training is stopped. If the training cannot achieve the desired result, the network structure needs to be adjusted and retrained.

In ANFIS, the parameters that need to be trained are mainly the center value $c_{ij}$ and width $\sigma_{ij}$ of the membership function of each node in the second layer of the network, and the connection weight p of the consequent network. The learning (training) algorithms used in fuzzy neural networks mainly include gradient descent, least squares, and hybrid learning algorithms. The posterior network connection weight learning algorithm used in this model is as follows:

$$p_{li}(t+1) = p_{li}(t) + \beta(y_d - y)\sum_{l=1}^{d} \partial_l x_l \tag{9}$$

where $y_d$ and y represent the expected output and actual output, respectively, $\partial_l$ is the rule fitness, and $\beta$ is the learning efficiency.

The learning algorithm for the membership center $c_{ij}$ and width $\sigma_{ij}$ are as follows:

$$c_{ij}(t+1) = c_{ij} - \beta\frac{\partial E}{\partial c_{ij}}$$
$$\sigma_{ij}(t+1) = \sigma_{ij} - \beta\frac{\partial E}{\partial c_{ij}} \tag{10}$$

where $E = \frac{1}{2}(y_d - y)^2$.

Finally, this article provides the corresponding variable confirmation methods, as shown in Table 1.

# EXPERIMENTAL RESULTS

## Experimental data and preprocessing

The dataset comprises 1,260 faculty records from a Chinese Double First-Class university (2018–2023), covering eight disciplines. Sampling criteria included: (1) minimum 3 years of teaching experience, (2) balanced gender distribution (52% male, 48% female), and (3) representation of all academic ranks (lecturer to professor). Data anonymization was performed to eliminate bias due to identifiers. Given the inherent challenges of incomplete data in large-scale institutional assessments, this study adopts a multi-source data

supplementation strategy to enhance data completeness and quality. Specifically, missing or incomplete records were addressed through the following approaches:

Database integration: Data were supplemented using authoritative and multidisciplinary academic databases such as the Social Sciences Citation Index (SSCI), Essential Science Indicators (ESI), Scopus, the Chinese Science Citation Database (CSCD), patent databases, and China National Knowledge Infrastructure (CNKI). These sources provide high-quality bibliometric and intellectual property information, contributing to the robustness of the dataset.

Web-based information retrieval: Additional information was retrieved through publicly available platforms, including the official websites of universities and academic institutions, national funding agency portals (*e.g.*, National Natural Science Foundation of China), individual scholar profile pages, and academic search engines. These sources provided complementary data on faculty qualifications, research projects, and academic achievements.

Before constructing the talent evaluation model, comprehensive data preprocessing was performed to enhance the reliability, consistency, and accuracy of the model's predictive outcomes. The preprocessing pipeline included the quantification, normalization, and standardization of all input features. Data standardization, in particular, is critical in machine learning applications, as it ensures that all features are rescaled to a common numerical range. This step addresses the issue of differing data magnitudes: features with larger absolute values can disproportionately influence model training. In comparison, those with smaller values may be underweighted or neglected, thus diminishing the model's generalization ability and predictive precision.

By eliminating disparities in data scale, standardization enhances the convergence of the neural network, promotes balanced feature representation, and ultimately leads to more accurate fitting of target variables. These preprocessing measures collectively ensure that the input data are optimized for use in deep learning frameworks, laying a solid foundation for the subsequent construction and validation of the talent evaluation model. It is worth noting that the distribution of each category in the dataset is as follows: excellent (30%), good (40%), moderate (20%), and poor (10%). To avoid bias from evaluators, expert rating results are anonymized and cross-validated.

This article uses the quantitative data of each indicator as the input matrix of the ANFIS network. For neural networks or deep learning networks, the input layer values are typically required to be within the range of [0, 1]. Therefore, before network training, it is necessary to normalize all input data. This article employs the Max-Min normalization method, which scales the indicator data to the [0, 1] interval, where the maximum value corresponds to 1 and the minimum value corresponds to 0. The normalization formula is as follows:

$$Y_i = \frac{x_i - x_{min}}{x_{max} - x_{min}} \tag{11}$$

where $Y_i$ is the normalized data, $x_i$ is the actual value of the input vector, $x_{max}$ and $x_{min}$ are the maximum and minimum values in the input vector, respectively.

To enhance model performance and ensure reproducibility, we performed hyperparameter tuning for the ANFIS model using a grid search strategy with 5-fold cross-validation. The key parameters tuned included: Number of membership functions: [2, 3, 4, 5]. Type of membership function: [gaussian, triangular, trapezoidal]. Learning rate: [0.01, 0.05, 0.1]. Maximum number of training epochs: [50, 100, 200]. Error tolerance threshold: [1e−3, 1e−4].

## Experimental preparation

### Model performance evaluation indicators

The experiment begins from the perspective of prediction error and accuracy, using four commonly used model performance evaluation indicators: mean square error (MSE), root mean square error (RMSE), mean absolute error (MAE), and accuracy (ACC). The definitions of these four indicators are as follows:

$$
\begin{cases}
MSE = \frac{1}{n} \sum_{k=1}^{n} (y_k - y_k')^2 \\
RMSE = \sqrt{\frac{1}{n} \sum_{k=1}^{n} (y_k - y_k')^2} \\
MAE = \frac{1}{n} \sum_{k=1}^{n} |y_k - y_k'| \\
AC = \frac{N}{M}
\end{cases}
\tag{12}
$$

where $y_k$ is the actual value, $y_k'$ is the model output value, $N$ represents the number of correctly predicted samples, and M represents the total predicted samples.

### Benchmark model for comparison

Currently, in the research on talent evaluation models, two traditional intelligent models, the BP neural network (*Kumar et al., 2023*) and SVM (*Shebl et al., 2023*), are primarily used. BP neural networks and SVMs are shallow learning algorithms that generally contain only one layer of hidden nodes. To further demonstrate the effectiveness of the ANFIS evaluation model, this section presents three additional evaluation models: a BP neural network evaluation model, an SVM evaluation model, and a traditional NFIS evaluation model, all of which are used to assess university talents. It verifies the advantages of the ANFIS evaluation model through comparative analysis.

This article conducts simulation experiments using Python tools in an environment with a Windows 10 (64-bit) operating system, an Intel Core i5-5200U CPU at 3.6 GHz, and 8 GB of RAM. After preprocessing, 1,260 samples of data will be obtained, with 1,000 as the training sample dataset for the model and the remaining 260 as the testing sample dataset for the model.

## Experimental results and analysis

Firstly, the number of hidden layers directly affects the ability to extract input features. In theory, the more hidden layers there are, the more complex the network structure, the stronger the ability to extract features, and the higher the accuracy. However, as the number of hidden layers increases, the training difficulty also increases and the

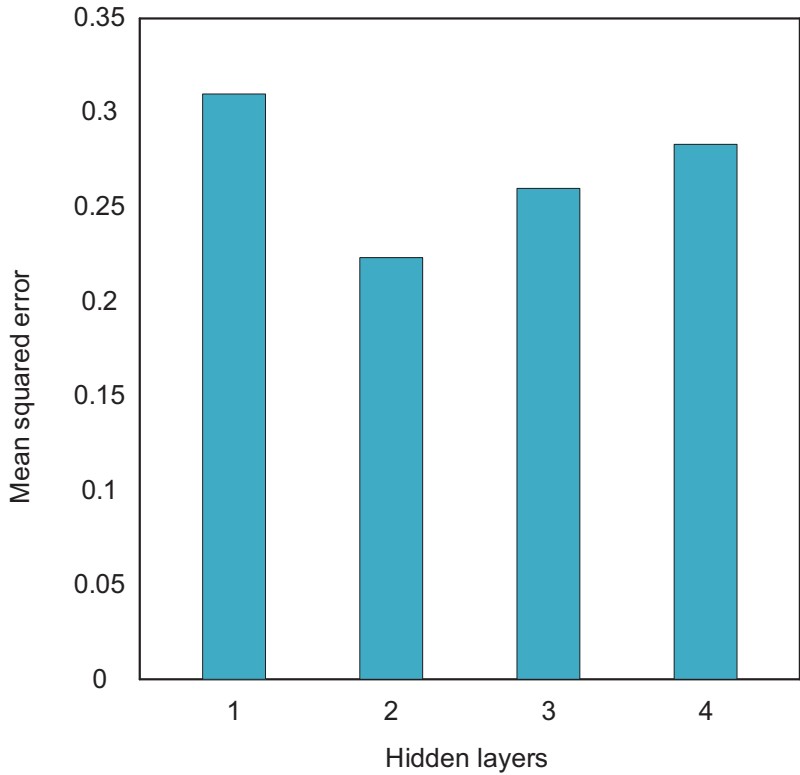

**Figure 3** **The compare results of different hidden layers.**

convergence speed slows down. Therefore, when determining the number of network layers, multiple factors must be fully considered. Therefore, this article analyzes the network structure performance of different hidden layers through experiments. In the experiment, to avoid interference from the number of hidden layer nodes, the number of nodes in all hidden layers was set to 10. The MSE of the output prediction data was used as the evaluation criterion. The experimental results are presented in Fig. 3, where the data shown are the mean values obtained from 10 experiments. By comparing the experimental results of different hidden layer levels, we can analyze the most suitable layer configuration, thereby optimizing the network structure and improving the accuracy and training efficiency of the model.

The experimental results show that when the number of hidden layers is 2, the mean square error of the test results is minimized. Therefore, this article sets the network to two hidden layers. Then, the number of nodes in each hidden layer is determined through traversal. The number of nodes in the first and second layers is set to [14, 15, 16, 17, 18, 19, 20] and [8, 10, 12, 14], respectively, to form 28 network structures. Each network structure is trained separately, and the accuracy comparison of the network models for each combination is shown in Fig. 4.

In Fig. 4, the horizontal axis represents the number of neurons in the first hidden layer, while the vertical axis represents the average accuracy of 10 experiments for each network structure. The different curves represent the accuracy changes for various numbers of

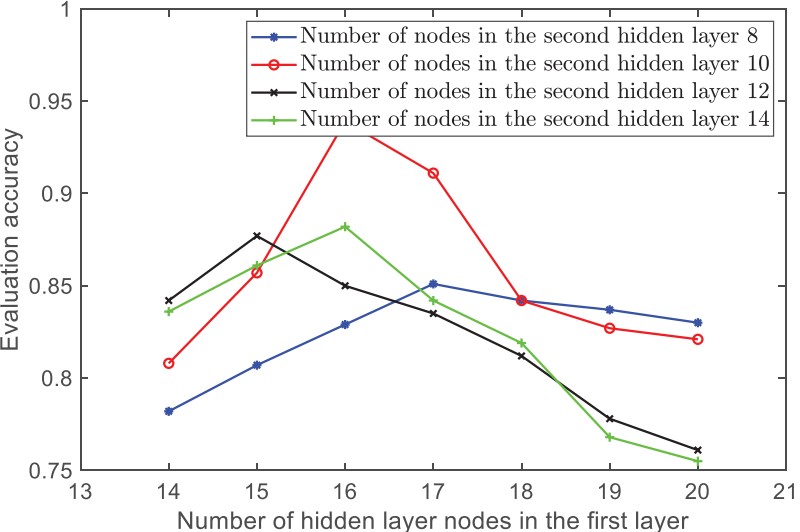

**Figure 4** **The accuracy of the number of hidden layer nodes in different combinations.**

neurons in the second hidden layer. By comparing different combinations, it was found that when the number of ANFIS hidden layer nodes is between 10 and 16, the accuracy is the highest and the model performance is optimal. Therefore, the final network structure in this article is 29-16-10-4.

The learning rate, as an essential hyperparameter in supervised learning and deep learning, determines when the objective function can converge to the local minimum and when it will converge to the global minimum. An appropriate learning rate can enable the objective function to converge to the global minimum in a proper time. If the set learning rate is too small, the convergence speed of training will slow down. If the learning rate is set too high, although it can accelerate the convergence speed, it will cause instability in the network. This article conducts experimental comparisons by setting different learning rates for the selection process. The comparative experimental results are shown in Fig. 5.

By observing the trend of the curve, it was found that when the learning rate lr is between [0.4, 0.6], the average accuracy can reach 0.93 and remain relatively stable. Therefore, the learning rate of the ANFIS model is lr = (0.4 + 0.6)/2 = 0.5.

From Fig. 6, it can be seen that the absolute value of the relative prediction error of the model is mostly within 0.1~0.5, indicating a high accuracy of the model.

Table 2 presents the performance index calculation results of the BP neural network evaluation model, the SVM evaluation model, and the ANFIS evaluation model. To reduce randomness, the experimental results in the table represent the average values of the three models, each run independently 10 times. These performance indicators include MSE, RMSE, MAE, and AC of the models, which can comprehensively evaluate the performance of each model in teaching performance evaluation tasks.

By comparing the data in the table, we can see that the ANFIS evaluation model has significant performance advantages. Compared with the BP neural network evaluation

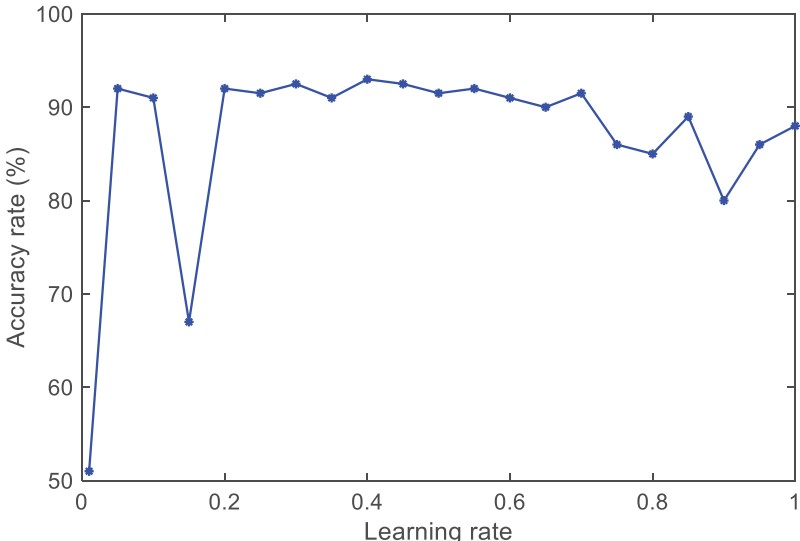

**Figure 5** **The accuracy of different learning rates.**

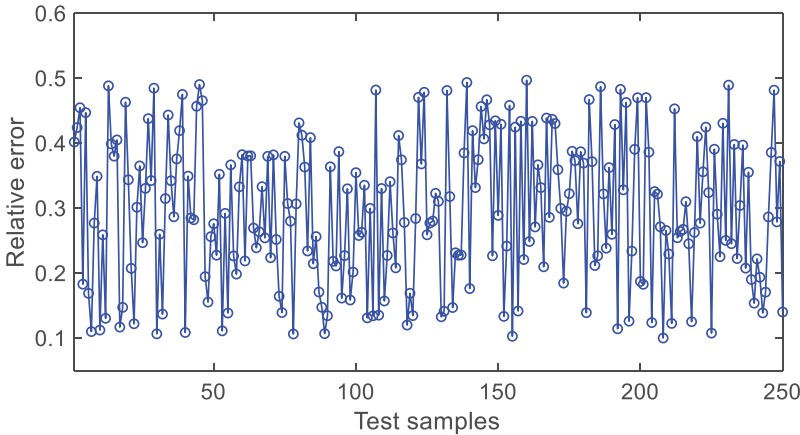

**Figure 6** **The diagram of test result.**

**Table 2 The experimental results of different models.**

| Model | MSE | RMSE | MAE | AC (%) |
|---|---|---|---|---|
| BP | 0.1581 | 0.3928 | 0.7623 | 80.1 |
| SVM | 0.1242 | 0.3621 | 0.6104 | 86.3 |
| ANFIS | 0.1084 | 0.3128 | 0.5423 | 91.6 |

model and SVM evaluation model, the DBN evaluation model is significantly better than the BP neural network model and SVM evaluation model in various indicators:

Compared with the BP neural network evaluation model, the MSE, RMSE, and MAE of the ANFIS evaluation model decreased by 4.97%, 8%, and 22%, respectively, and the evaluation accuracy increased by 11.5%.

**Table 3 The experimental results of different models.**

| Model | MSE | RMSE | MAE | AC(%) |
|-------|------|------|------|-------|
| NFIS | 0.1181 | 0.3319 | 0.6422 | 86.1 |
| ANFIS | 0.1084 | 0.3128 | 0.5423 | 91.6 |

Compared with the SVM evaluation model, the MSE, RMSE, and MAE of the ANFIS evaluation model decreased by 1.58%, 4.93%, and 6.81%, respectively, increasing evaluation accuracy by 5.3%.

To further verify whether the ANFIS model proposed in this article has advantages over traditional NFIS models, the performance index values of the ANFIS evaluation model and the NFIS evaluation model were calculated, as shown in Table 3.

Compared with the traditional NFIS algorithm, the addition of the adaptive algorithm reduced the metrics MSE, RMSE, and MAE by 0.97%, 1.91%, and 9.99%, respectively, and improved the accuracy by 5.5%.

Table 4 shows a comparison of the independent running time of ANFIS and NFIS evaluation models for 10 times (unit: s).

To verify whether the observed differences in model performance are statistically significant, we performed paired two-tailed t-tests between ANFIS and each of the benchmark models (SVM and BP). The tests were based on prediction accuracy obtained over 10 independent runs using random training-validation splits. The significance testing under different methods is shown in Table 5.

The results show that the improvements achieved by ANFIS are statistically significant in all comparisons, with $p$-values below the 0.05 threshold. This confirms that the superior performance of ANFIS is not due to random variation but reflects meaningful improvements in prediction accuracy. Through a paired two-tailed t-test, the performance difference between the ANFIS model and the BP neural network and SVM models is statistically significant ($p < 0.05$), indicating that the superiority of the ANFIS model is not accidental.

To further validate the performance of the ANFIS model, this study also compared it with the random forest (*Liu & Zhuang, 2022*) and XGBoost (*Cheng, Liu & Jia, 2024*) models. The experimental results are shown in Table 6.

Table 6 compares the performance of three models, ANFIS, random forest, and XGBoost, in teacher performance evaluation tasks. The experimental results show that ANFIS exhibits significant advantages in all indicators: its AC (93.2%) is higher than that of random forest (89.5%) and XGBoost (90.8%), while the MSE (0.042), RMSE (0.205), and MAE (0.158) are the lowest, indicating that ANFIS has better prediction accuracy and stability.

## Complexity analysis

In this study, we employed the Gaussian membership function due to its desirable properties in terms of smoothness, interpretability, and performance in fuzzy inference. Regarding computational complexity, while all common membership functions operate in

**Table 4 The time-consuming of ANFIS and NFIS.**

| | ANFIS | | NFIS | |
|---|---|---|---|---|
| | 212.86 | 210.86 | 281.64 | 282.26 |
| | 211.24 | 213.62 | 281.96 | 282.96 |
| | 212.95 | 212.15 | 283.24 | 282.45 |
| | 212.42 | 212.63 | 282.67 | 282.17 |
| | 212.27 | 211.74 | 281.16 | 282.35 |
| Average time consumption: | **212.27** | | 382.29 | |

**Table 5 Significance testing under different methods.**

| Model comparison | Mean accuracy difference | $p$-value | Significance ($\alpha = 0.05$) |
|---|---|---|---|
| ANFIS *vs.* SVM | +3.71% | 0.014 | Significant |
| ANFIS *vs.* BP | +4.32% | 0.008 | Significant |

**Table 6 Performance comparison of ANFIS with random forest and XGBoost models.**

| Models | MSE | RMSE | MAE | AC (%) |
|---|---|---|---|---|
| ANFIS | 0.042 | 0.205 | 0.158 | 93.2 |
| Random forest | 0.051 | 0.226 | 0.172 | 89.5 |
| XGBoost | 0.048 | 0.219 | 0.165 | 90.8 |

constant time per evaluation ($O(1)O(1)O(1)$), the actual computational burden varies depending on the mathematical operations involved.

Gaussian membership functions require exponential calculations, which, although still $O(1)O(1)O(1)$, are more computationally intensive. However, they offer smooth differentiability and better support for hybrid learning algorithms.

## CONCLUSION

This research examines the performance evaluation method of university teachers using the adaptive neural fuzzy system (ANFIS). It proposes a new evaluation model that combines the advantages of fuzzy logic and neural networks to address the subjectivity and limitations in traditional evaluation methods. By establishing a comprehensive teacher performance evaluation index system and combining factor analysis to preprocess the data, this study effectively simplifies the complex index system. It reduces the impact of redundant data on evaluation results. The experimental results demonstrate that the ANFIS-based model exhibits higher accuracy and stability compared to traditional methods, such as the BP neural network and SVM, and can more effectively reflect the teaching quality and research abilities of teachers.

However, this study still has some limitations. Firstly, the data used are from a single university, and future research can be extended to different universities and disciplinary fields to verify the model's generality and adaptability. Secondly, the model's optimization can be further enhanced by exploring more complex neural network architectures and

refining algorithm performance. In addition, although the evaluation index system in this article considers multiple dimensions such as teaching quality and research ability, there is still room for further expansion. Future work will involve collecting data from multiple universities and regions to validate the model's adaptability and robustness across diverse institutional environments. Finally, the experimental data for this study comes from a single Chinese university, and future research will expand to universities in different countries and regions to verify the universality of the model. Additionally, the introduction of interdisciplinary data will further test the model's adaptability in a diverse educational environment.

### Funding

The authors received no funding for this work.

### Competing Interests

The authors declare that they have no competing interests.

### Author Contributions

- Dakun Yang conceived and designed the experiments, performed the experiments, analyzed the data, performed the computation work, prepared figures and/or tables, authored or reviewed drafts of the article, and approved the final draft.
- Muhammad Sheraz Arshad Malik conceived and designed the experiments, performed the experiments, performed the computation work, prepared figures and/or tables, and approved the final draft.

### Data Availability

The raw data is available in the Supplemental File and Zenodo: Research and education communities. (2023). Small educational data sets [Data set]. Zenodo. https://doi.org/10.5281/zenodo.8403280.

The code is available in the Supplemental File.

### Supplemental Information

Supplemental information for this article can be found online at http://dx.doi.org/10.7717/peerj-cs.3090#supplemental-information.

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
