# Peer review of "Design of performance evaluation method for higher education reform based on adaptive fuzzy algorithm"

_PeerJ Computer Science, doi:10.7717/peerj-cs.3090_

## Round 0.1 · original submission · Major Revisions

Reviewer 1 ·

Basic reporting

The English language needs editing for good perception. It is advisable to give more thorough explanations when using abbreviations, etc.

The sources used can be improved, based on an analysis of the world experience of journals included in Scopus. It is advisable to include an analysis of various rating systems for measuring the effectiveness of teachers and the level of talent measurement by traditional means, of leading universities in the world, in the analysis of world experience. Also, the article does not provide comparative analyses of other scientific methods of machine learning to solve this problem; expanding the description will allow readers to better understand the contribution of the study to world research in general.

The structure of the article and tables meet the requirements of the journal. However, it is desirable to provide more illustrative materials of good quality.

The hypotheses put forward are tested, but, unfortunately, the exact amount of processed material and the conditions for forming a sample for the evidence base are not provided.

The official results contain sufficiently clear definitions of terms and relevant evidence.

Experimental design

The research is conducted within the framework of the goals and objectives of the journal. The research question is clearly defined, relevant, and meaningful.

The solution developed by the authors is presented at a high technical level and meets the standards.
The methods are described in sufficient detail and information for repetition.

Validity of the findings

The novelty of the study is present, but it requires substantiation of the use of the presented research methods. The development of software for analyzing the rating of teachers is a rather urgent problem that needs to be solved. The conclusions presented in the article confirm the results of the study and are reflected in the content of the article.

Additional comments

The authors conducted a thorough study, but some suggestions have the potential to improve it, namely: the content analysis of previous studies and the study of world experience were not sufficiently thorough; clear conditions and the volume of the experimental sample are not provided, which is desirable for confirming the hypothesis, its selection and formation need to be clarified, which will allow understanding its quality and relevance; The choice of development language is not sufficiently justified; the content of the article does not fully reflect the stages and interaction of university administrations regarding the use of the results of this study; additional recommendations and screenshots may be needed.

·

Basic reporting

- The introduction should provide a clearer justification for why an Adaptive Neuro-Fuzzy Inference System (ANFIS) is preferable to other methods beyond accuracy improvements.
- More recent references (2024 or 2025) should be included.

Experimental design

- While the methodology describes the ANFIS model in detail, additional explanation on how hyperparameters were chosen for model training could enhance reproducibility.
- The description of factor analysis could be expanded to clarify how the selection of sub-indicators was made and why these specific indicators were chosen.

Validity of the findings

- The study could benefit from a statistical significance test to determine whether the performance improvements of ANFIS over other methods are statistically significant.
- The paper mentions optimization of training data but does not elaborate on potential overfitting issues.

Reviewer 3 ·

Basic reporting

The paper is structured, the approach is logical, the language is clear, and the flow is maintained.
Some figures have missing labels. Minor typo errors are observed.

Experimental design

1. My first concern is about the sample. More heterogeneous DMUs could make it a bit generalized.
2. Why have the authors considered the suggested membership function? Please provide a comparative study with other membership functions.
3. Justify the selection of a value for the parameter. What would happen to other possible values?

Validity of the findings

I would like to see statistical tests to validate the results.

Additional comments

The authors must discuss the computational complexity.

---

## Round 0.2 · Minor Revisions

Dear Dr. Yang and Dr. Malik,

Thank you for submitting your manuscript entitled "Design of a performance evaluation method for higher education reform based on an adaptive fuzzy algorithm" to PeerJ Computer Science. Based on the reviewers' assessments, we are pleased to inform you that the manuscript is acceptable for publication pending minor revisions.

Both reviewers recognize the value of your contribution, particularly the innovative use of ANFIS for faculty performance evaluation. Reviewer 4 highlighted the clarity of your experimental design and the model's improved accuracy over traditional methods. Reviewer 1 emphasized the need for a more comprehensive literature review and better contextualization within global research efforts.

To strengthen the manuscript, please clarify the rationale for using EFA and its integration with ANFIS, expand comparisons with modern machine learning techniques, and provide clearer explanations of key equations and figures. Enhancing the language, justifying methodological choices, and detailing your dataset and experimental sample will also improve clarity and reproducibility.

We look forward to receiving your revised manuscript.

Reviewer 1 ·

Basic reporting

Requires refinement, based on the previous round of review.

Experimental design

Requires refinement, based on the previous round of review.

Validity of the findings

Requires refinement, based on the previous round of review.

Additional comments

Requires refinement, based on the previous round of review.

Reviewer 4 ·

Basic reporting

The article proposes a university faculty performance evaluation model based on an Adaptive Neuro-Fuzzy Inference System (ANFIS), which combines fuzzy logic and neural networks to achieve more accurate and objective assessments. Starting with a system of 16 indicators, factor analysis is applied to reduce redundancy and simplify the framework. The model is trained using real data and fine-tuned through hybrid algorithms, showing significant superiority over traditional methods such as BP neural networks and SVM. The results highlight its higher accuracy, interpretability, and practical value for academic management.

This is an interesting and timely study that addresses a relevant challenge in the field of higher education management—namely, the objective evaluation of university teaching performance. By integrating fuzzy logic and neural networks within an Adaptive Neuro-Fuzzy Inference System (ANFIS), the authors provide a novel approach that effectively handles the complexity and uncertainty inherent in educational assessment.

Among the strengths of the manuscript are its comprehensive and well-structured evaluation index system, the use of robust data from a reputable institution, and the methodical application of factor analysis to improve model efficiency. The experimental design is clearly explained, and the comparative analysis with traditional models (BP and SVM) convincingly demonstrates the advantages of the proposed framework in terms of accuracy and practical applicability. Overall, the study makes a valuable contribution to the ongoing efforts to enhance data-driven decision-making in academic settings.

While the manuscript presents a promising and innovative evaluation model, several areas could benefit from clarification and improvement. First, the use of figures and tables is not always accompanied by adequate references or explanatory text within the manuscript, which limits their interpretability for the reader. In addition, while the use of Exploratory Factor Analysis (EFA) is appropriate, the rationale for selecting this method is not fully justified, nor is its connection to the subsequent ANFIS architecture clearly established.

Moreover, several mathematical expressions—such as equations (1) and (2)—are introduced without a complete explanation of their components or relevance, which may hinder understanding for readers unfamiliar with the specific techniques. In terms of model comparison, the study includes only classical baselines (SVM and BP), without considering more recent and powerful methods such as Random Forest, XGBoost, or deep learning classifiers. This omission limits the scope of the performance evaluation. Lastly, although the authors acknowledge that the data come from a single Chinese university, a more explicit discussion on how the model could generalize to international or multidisciplinary contexts would strengthen the study. Additional information on the dataset—such as class distribution, source diversity, and possible evaluator bias—would further enhance transparency and replicability.

Experimental design

Please refer to my comments above

Validity of the findings

Please refer to my comments above

Additional comments

Please refer to my comments above

---

## Round 0.3 · accepted · Accept

I think that the paper has now properly addressed the issues raised by the reviewers. Congratulations